# Procedures for Building a Secure Environment in IoT Networks Using the LoRa Interface

**DOI:** 10.3390/s25133881

**Published:** 2025-06-22

**Authors:** Sebastian Łeska, Janusz Furtak

**Affiliations:** Faculty of Cybernetics, Military University of Technology, 00-908 Warsaw, Poland; janusz.furtak@wat.edu.pl

**Keywords:** security in IoT, IoT authentication, trusted IoT environment, trusted platform module, PCR registers

## Abstract

**Highlights:**

**What are the main findings?**
IoT devices have limited resources for performing cryptographic operations and utilize low-bandwidth wireless links to facilitate long-distance data exchange.There is a need to develop appropriate authentication and symmetric key distribution mechanisms for constrained IoT devices to ensure network security.

**What is the implication of the main finding?**
A set of procedures has been developed to build a trusted IoT network in an uncontrolled and untrusted environment using a novel LoRa communication scheme.A novel authentication mechanism has been developed for IoT devices using a hardware TPM and PCR registers.

**Abstract:**

IoT devices typically have limited memory resources and computing power. For this reason, it is often not possible to use the authentication and trusted environment mechanisms commonly used on the Internet. Due to the autonomous operation of IoT devices, solutions that require user interaction should be excluded. Additionally, due to the limited capabilities of IoT devices, mechanisms performing complex cryptographic operations are also not always recommended. This paper proposes a set of mechanisms for building a trusted IoT environment using a hardware TPM 2.0 module. The developed set includes procedures for securely registering nodes in the network, which are designed for use in an untrusted and uncontrolled environment. The authors also proposed a protocol for device authentication using PCR registries supported by the TPM based on the Proof of Knowledge concept. Using a direct method, the solution also involves implementing a symmetric key distribution protocol based on the KTC (Key Translation Centre) scheme. The developed procedures can be used in networks where nodes have limited memory resources and low computing power. The communication interface used in the developed demonstrator is LoRa (Long Range), for which a proprietary method of identifying network devices has been proposed to ensure the confidentiality of the communicating parties’ identities.

## 1. Introduction

In the era of rapidly developing new technologies, there are instances where security issues are overlooked when introducing new solutions. This problem often applies to IoT networks [1]. Due to low computing power and limited memory resources, IoT devices often lack built-in cryptographic procedures, which exposes them to numerous cyberattacks [2,3]. Addressing this problem requires developing a set of mechanisms that will provide an adequate level of security regardless of the existing limitations of IoT devices.

On the Internet, asymmetric cryptography is commonly used to build security structures. This approach enables the easy exchange of encryption keys and authentication, ensuring data trustworthiness through digital signatures. Unfortunately, this solution is often ineffective for IoT networks due to the required computing power, memory resources, and available data transmission medium.

Symmetric cryptography puts less strain on the hardware resources of IoT devices than asymmetric cryptography while providing the same level of security. AES-128 encryption is estimated to provide a similar level of security to RSA-3072 [4]. Additionally, using shorter cryptographic keys requires less computing power and time to encrypt data. An important aspect that also needs to be considered is the low resistance of asymmetric cryptography to quantum attacks, in part due to the possibility of utilizing the Shor algorithm [5].

While symmetric cryptography is generally considered more resilient to quantum attacks than asymmetric schemes, it is not entirely immune. In particular, Grover’s algorithm [6] enables a quadratic speed-up for brute-force key search, effectively reducing the security of an n-bit symmetric key to approximately n⁄2 bits. It means that a 128-bit key provides roughly 64 bits of post-quantum security, which is below the commonly recommended threshold for long-term data protection. Therefore, to maintain an adequate security margin in the presence of potential quantum adversaries, the use of 256-bit symmetric keys (e.g., AES-256) is recommended, offering an effective quantum security level of 128 bits. Additionally, key lifetimes in such systems should be minimized, with session keys refreshed frequently and long-term keys rotated based on system-specific threat models.

To ensure a sufficient security margin, especially in the context of quantum threats such as Grover’s algorithm, the lifetime of session encryption keys should be strictly limited. According to the guidelines provided in NIST SP 800-57 Part 1 Rev. 5 [4], encryption keys used to secure large volumes of data over short periods (e.g., for link encryption) should have an originator usage period on the order of a day or a week, while keys used for smaller volumes may be valid for up to two years. In the context of quantum threats such as Grover’s algorithm, similar key lifetimes may still be appropriate, provided that the symmetric key length is doubled (e.g., using AES-256 instead of AES-128) to maintain an equivalent security level.

However, when symmetric cryptography is used, a significant problem arises: key distribution. Since a single key is used for encryption and decryption of data, it is necessary to transmit this key between devices. In such a situation, it is possible to use key establishment algorithms such as the Diffie-Hellman protocol [7] or ECDH [8] to establish a shared symmetric key over an unsecured transmission medium. However, this raises the problem of the lack of a trusted third party to confirm the identity of both nodes.

It is also common to use symmetric key distribution protocols, which involve generating a key on one side and sending it securely to the other devices on the network. This solution is flexible because it is not limited to a pair of nodes. It is possible to distribute such a key to any number of network devices. However, these protocols require a central node to oversee the distribution of such keys and confirm their trustworthiness. There is currently a lot of research into key distribution mechanisms. Proposed solutions are based on the physical layer of devices [9], node attributes [10], or the transmission of ready-made keys generated using a strong entropy source [11,12]. Unlike key establishment protocols, each of these solutions offers the flexibility to distribute a single key to either a pair of nodes or multiple nodes in the network.

Some research in building a trusted IoT environment also focuses on Distributed Ledger Technology (DLT) [13,14,15]. In this approach, all parties using DLT within one or more organizations keep copies of the data in their resources and can update them according to agreed-upon rules. The implementation of this technology in IoT networks ensures data integrity and immutability. With DLT, it is also possible to securely support M2M (Machine-to-Machine) transactions and smart contracts. Each device in the DLT network can add entries to the register in an immutable and verifiable way, which is crucial in Industrial IoT (IIoT) networks.

Another approach to building a trusted IoT environment is the centralized approach [16,17,18,19]. In a centralized IoT system architecture, management, data processing, and access control are performed using a central network node. Such a structure is easier to manage; however, it requires the implementation of appropriate procedures to ensure the security, integrity, and control of IoT devices. Centralized networks require centralized identity management, where each device possesses a unique identity and authentication certificate. The central node must also control network access for the remaining devices and supervise which devices are allowed to communicate with each other. In such networks, encryption key management is often handled centrally as well.

In recent years, significant progress has been made in the field of applied cryptography, particularly in the design of lightweight and chaotic encryption algorithms for data protection in constrained environments. Novel schemes have been proposed for secure image processing, such as the use of high-quality reconstruction with optimized orthogonal compressive sensing and nonlinear dynamics [20], hybrid chaotic encryption with cellular automata and 4D modulation tailored for data center environments [21], or the application of fractal sorting matrices and Fibonacci Q-matrices for privacy-preserving image security [22]. Additionally, approaches combining compound dynamic diffusion and efficient single-round encryption based on 4D nonlinear dynamical systems have demonstrated promising results in terms of both speed and robustness [23]. Although these solutions primarily target multimedia privacy, they highlight current trends in cryptographic system design, including high efficiency, resistance to differential attacks, and adaptability to constrained platforms such as IoT devices or embedded systems. The trust establishment protocol presented in this work follows the same design philosophy, prioritizing minimal resource consumption while ensuring resilience against impersonation and key compromise.

Although the primary focus of this work is to propose a trust-based key distribution and authentication mechanism tailored to constrained Internet of Everything (IoE) environments, it is essential to position the solution against existing protocols, such as OSCORE [24] and DTLS [25]. Both OSCORE and DTLS are widely adopted in IoT ecosystems and offer standardized security mechanisms. However, they rely on pre-established trust or PKI infrastructure and are not optimized for scenarios involving dynamic, ad hoc node registration without prior credential distribution. Moreover, the proposed protocol introduces minimal message overhead (a single message broadcast during the registration phase) and reduces the per-transaction cryptographic load by avoiding full TLS/DTLS handshakes. In contrast to OSCORE, which focuses on application-layer security and requires CoAP integration, our approach can operate independently of the application protocol stack. While a detailed benchmarking study (e.g., energy consumption or memory usage) is outside the scope of this paper, the proposed method provides clear advantages in scenarios where secure initialization and identity establishment must occur without preconfigured secrets or continuous internet access.

An essential issue in IoT networks is ensuring trust between network nodes. In traditional computer networks, this can be achieved using authentication mechanisms or certificates signed and issued by certification authorities [26]. In IoT networks, nodes typically do not have direct access to the Internet, which means that a trusted third party cannot verify their identity. In such a situation, it is necessary to implement authentication mechanisms that ensure the trustworthiness of the devices, are lightweight and efficient, and prevent unwanted users from joining the network. Solutions based on the use of biometrics and mathematical operations [27], authentication through the ability to encrypt data [28], password knowledge and hash functions [29], or using a RADIUS server [30] are proposed in this regard.

One of the key challenges in developing secure IoT environments is the lack of mechanisms that enable the establishment of initial trust between devices without prior knowledge of their security parameters. It creates a potential risk of introducing malicious devices that aim to intercept or collect sensitive data. Furthermore, many existing solutions are not optimized for resource-constrained devices or require continuous Internet connectivity, which limits their applicability in mobile or remote scenarios. Some approaches assume operation in fully controlled environments, a condition that is often unrealistic in practical deployments. Additionally, most existing works fail to provide a comprehensive trust management framework, often omitting critical procedures such as node registration, authentication, and trust propagation. As a result, available solutions typically address only selected aspects of the trust establishment process. The solution proposed in this paper aims to overcome these limitations by providing a comprehensive and lightweight framework that supports all key trust management procedures and is capable of operating in uncontrolled environments and on devices with limited computational and communication capabilities.

The solution proposed in this paper encompasses a set of mechanisms for establishing a trusted environment for IoT networks, where communication occurs over a LoRa interface. The developed procedures are designed to work in an untrusted environment and have been adapted to build trust from the outset of network operation. It is impossible to compromise the transmitted data or plant unauthorized devices. The presented set of mechanisms includes a proprietary method for authenticating IoT nodes, a procedure for registering new devices, and a method for distributing symmetric cryptographic keys. In addition, a novel method for identifying network devices has been proposed, which makes it impossible for unauthorized parties to identify the sender and receiver of a message.

This paper focuses on the design and analysis of network-level procedures related to trust establishment, node registration, and mutual authentication in IoT environments. Therefore, we consider a network adversary model in which the attacker can eavesdrop on communication, attempt message injection or replay and try to impersonate a legitimate node. In this context, the proposed set of procedures provides protection against impersonation, unauthorized access, and protocol-level forgery. Physical attacks, including direct tampering with hardware, modification of firmware, or side-channel attacks targeting the TPM, are considered out of scope, as this work does not address device-level hardening. We assume the integrity of the TPM and its secure execution environment, in line with standard trust models adopted in similar work.

The remainder of the article is organized as follows. Section 2 presents the concept of the developed procedures. Section 3 describes the developed system demonstrator and presents the system results. Development plans are included in Section 4.

## 2. Procedures for Building a Trusted Environment

IoT devices tend to be large and mobile, which means low-power wireless interfaces are needed. Such links have a low data throughput. For these reasons, the security mechanisms used should be lightweight and efficient.

Building a trusted environment requires the implementation of security mechanisms that operate from the beginning of the network and work continuously. To this end, it is necessary to implement procedures responsible for registering nodes on the network, properly authenticating those nodes, ensuring trustworthiness, and encrypting data exchange.

The CE (Central Entity) will serve as the guarantor of trust and the party responsible for implementing these procedures. All other system nodes involved in the data exchange will be referred to as client nodes. The procedures described will include the following steps:Initializing an IoT node (including a CE node).Attaching the node to the network.Authentication of nodes with the central unit.Generation and distribution of session keys.Renewal of session keys.

Figure 1 shows the architecture of the system demonstrator, which assumes the presence of a CE node and *n* client devices (N_i_). Each device is assumed to be equipped with a TPM 2.0 (Trusted Platform Module) hardware module responsible for supporting cryptographic operations and a LoRa (Long Range) wireless communication module.

An essential feature of the designed system is the way devices are addressed. The LoRa interface operates only in the first layer of the ISO/OSI model [31], meaning it does not provide device addresses. In the proposed approach, device addresses are replaced by Session Identifiers (sIDs) and node Public Identifiers (pIDs).

sIDs in a four-byte number are allocated to each pair of devices between which data exchange is necessary. They only identify data transmission between these nodes (Figure 2). This identifier can only be known for the pair of nodes exchanging data. This approach allows the identity of the sender and receiver to be kept secret from unauthorized parties.

The pIDs of the nodes are known to all devices in the network and allow requests to the CE to be formulated to prepare data for the future establishment of secure sessions between two client nodes. Based on such a reference, it is possible to assign two devices to a common sID (Section 2.4).

In the remainder of the study, the following designations will be used:
SKSession Key (AES-256) generated by N_i_, dedicated to encrypting data transmitted between N_i_ and N_j_;RKRSA Key is intended to decrypt the received SK;RPK_Ni_RSA Public Key of the N_i_ node, used to encrypt the duplicate of SK;MK_Ni_Master Key (AES-256) of the N_i_ node and the CE node, established by entering the one-time key of the N_i_ node on the CE;LEKLocal Encryption Key (AES-256) of the node used to encrypt local data;AF_Ni_Authentication Factor (32 bytes) of the N_i_ node used in the authentication process;sIDconfidential Session ID (4 bytes) generated and assigned by the CE, used to identify data transmissions between a pair of nodes;pIDPublic ID assigned by the CE when a node joins the network, used to identify each node;pID_list_a list of public IDs of all network nodes with their descriptions;Rrandom number (4 bytes) generated to test the AF’s operation;S (sID, [M_i_, …, M_j_])content of the frame for sending the concatenation of the messages [M_i_, …, M_j_] to the node using sID;E (MK_Ni_, [M_i_, …, M_j_])encrypted form of the messages [M_i_, …, M_j_] concatenated with a master key MK_Ni_;BKBroadcast Key (AES-256) generated by CE to encrypt data transmitted from CE to all nodes.

### 2.1. Procedures for Initiating Nodes

The procedures developed involve performing an initial configuration on each client device at the production stage using the TPM 2.0 module. These activities are intended to prepare the device for joining the network and communicating securely within it.

The first step is for the TPM 2.0 module to generate a random number acting as a one-time key (OtK). This key will be forwarded to the CE for entry into the system (see Section 2.2 for a detailed description).

The next important step in initializing the node is to run the boot routine, which calculates the SHA-256 hash from hardware configuration data obtained from the **lshw** tool available in the Linux environment during boot. This tool provides data such as memory size, software version, motherboard configuration, processor version, and other relevant details. This hash is stored in one of the Platform Configuration Registers (PCRs) of the TPM and acts as an Authentication Factor (AF) (see Section 2.3). The system must always ensure the protection of the AF value during the node’s operation. The AF must not be disclosed to unauthorized parties. Making unauthorized modifications to a node’s configuration will result in a change to the AF value and, consequently, the inability of that node to authenticate itself to the network.

The next step is for the TPM to generate an RSA key pair, which will be used to distribute symmetric keys (see Section 2.4). These keys will be stored in the resources of the TPM. The device is ready for network operation once the above steps have been completed.

In the case of the CE node initialization, the above procedure is extended to generate a symmetric BK (AES-256), which will be used to encrypt the broadcast transmission.

### 2.2. Attaching a Node to the Network

To build a trusted environment, it is necessary to implement appropriate procedures from the beginning of the network. To this end, each node participating in communication should be registered with the CE.

A Ni node’s registration requires that the node’s OtK (see Section 2.1) is first entered in the CE node. This key to the resources of the CE node is manually entered by the admin before the registration of the Ni node starts (in the case of the creation of a new network comprising multiple nodes, the OtK insertion activity of these nodes can be performed in batch by placing these OtKs into a file and importing it into the CE node’s resources.). The OtK is used as a seed to generate an AES-256 symmetric key, also known as a Master Key (MK), for the node to be registered. The MK_Ni_ key is designed to secure data transmission between the Ni client node and the CE. The advantage of this approach is that this action nullifies the risk of substituting a false device. This approach gives the administrator complete control over which nodes join the network. This operation is also necessary because, without the administrator’s intervention, two devices previously unknown to each other cannot establish a secure connection without the risk of substituting another device, which could collect data from the network for an unauthorized party. The procedure for attaching an example node named N1 to the network is as follows (Figure 3):

Node N1, upon power-up, starts listening to the transmitted data.Node CE generates the key MK_N1_ based on the key OtK.Node CE generates for node N1:Session Identifier (sID)—a 4-byte unique identifier designed to identify the data transmission between a pair of devices (including the CE node) and known only to them;Public Identifier (pID)—known to all network devices.The CE node sends the N1 node the sID_N1_ assigned to it, its AF value, its RSA public key (RPK_CE_), and its BK key. The sID identifies the first sending of the message set to ‘0000’.Node N1 sends its AF value and its RPK_Ni_ public key to the CE node.The CE node sends the broadcast (sID set to ‘FFFF’) the current pID list (pID_list_) of all authorized devices belonging to the network, extended by node N1.

### 2.3. Authentication of Nodes with the CE Node

Building a trusted environment requires ensuring that the identity of each node in the network is trustworthy. To this end, a CE acts as a trusted third party.

Once each node has been registered, AF values are stored in the protected resources of the central unit. These values are used during authentication. The authentication mechanism developed belongs to the Proof of Knowledge group, which means that the authentication of each node is performed by proving knowledge of the AF values without revealing them. The authentication procedure presented below uses the following designations:

⨁—XOR operation;

+—concatenation;

AF_X_—AF value of node X stored in the PCR register;

H(M)—SHA-256 digest of M messages.

The steps of the authentication procedure between the client and the CE node:The client generates a 32-byte random number R using the local TPM.The client calculates the number C according to the following formula:(1)$$C={AF}_{CE}\;⨁{AF}_{N}⨁\;R+H({AF}_{N}⊕R)$$

3.The client sends the C-number to the CE node.4.The CE node identifies the sender from the sID and determines its AF_N_ based on its local data. Knowing the client’s AF value and AF_CE_, it calculates the R-number:


(2)
$$R={AF}_{CE}\;⨁{AF}_{N}⨁\;(C-H({AF}_{N}⊕R))$$


5.After calculating R, the CE node verifies its correctness and the correctness of the AF_N_ using the hash taken from the end of the C number.6.If the received data matches the digest, the CE node calculates P (SHA-256 digest) according to the following formula:


(3)
$$P=H({AF}_{CE}\;⨁\;R)$$


7.The CE node sends the P-number to the client.8.The client verifies the identity of the CE node by verifying the correctly calculated R.

The authentication mechanism developed is lightweight and does not require complex cryptographic operations. Despite this, it ensures the trustworthiness of the network nodes’ identities.

Each authentication factor is 32 bytes long, which means that the value space of such a factor is 32^256^ possible values. Its length is the same as that of the AES-256 key. Furthermore, the data transmitted over the network during authentication always consist of at least two 32-byte values, which no unauthorized party (either on or off the network) knows. In the worst-case scenario, attempting a brute-force attack involves reviewing $32^{256^{2}}$ values. The arguments cited above mean that it is impossible to forcibly obtain the Authentication Factor in a reasonable amount of time.

The value H(AF_N_ ⨁ R) is computed using a 32-byte secret factor assigned to the node (AF_N_) and a freshly generated 32-byte nonce R. Without knowledge of the AF_N_, an attacker is unable to compute the hash function correctly. Attempting to recover either AF_N_ or R via brute force would require searching for a 256-bit space, which is computationally infeasible with current technology. Therefore, successfully deriving these values based solely on the observed value C would imply breaking the preimage resistance of the SHA-256 function, which is widely considered infeasible under current crypto-graphic assumptions.

The protocol also ensures mutual authentication through a response from the Central Entity in the form of the value P = H(AF_CE_ ⨁ R), which is verified by the client. In this case, it is assumed that the value AF_CE_ is publicly known to all end nodes, meaning its secrecy is not required. The security of this step relies on the fact that only the CE knows the current nonce R (derived from the correctly processed challenge), and thus, the ability to generate a correct value P confirms the authenticity of the response. Therefore, the client does not verify the CE’s knowledge of a secret, but rather the CE’s ability to reconstruct R using the correct values AF_N_ and AF_CE_ and to process the challenge accordingly.

The correctness of the protocol is based on the following assumptions. First, SHA-256 is assumed to be resistant to preimage, second preimage, and collision attacks, in line with current cryptographic standards. Second, the random value R is generated using a cryptographically secure random number generator provided by the TPM, ensuring its unpredictability. Third, the value AF_N_ is never transmitted in plaintext over the network.

The introduction of sID and pID conceals session identifiers in the network, thereby enhancing the system’s resilience against attacks on LoRaWAN networks. However, some threats to LoRaWAN networks at the physical layer remain. Threats to LoRaWAN networks are considered against the following attacks [32]: sniffing, jamming, wormhole, key extraction, and energy attacks.

A successful sniffing attack, combined with security key extraction, can escalate to Man-in-the-Middle attacks and the creation of a covert channel, thereby compromising the physical layer. In the proposed solution, the introduction of the sID and pID, along with the authentication factor AF, which is determined by the current device configuration, eliminates these attacks. Additionally, the generated keys can be stored in secret form in the resources protected by the mechanisms offered by the TPM, which will prevent an adversary from extracting these keys.

Jamming and Energy attacks are difficult to thwart. The proposed solution does not support immunity against these attacks. In this respect, solutions based on radio-frequency fingerprinting are promising [33].

The base of wormhole attacks is an effective sniffing and jamming attack. In this respect, the proposed solution does not affect the system’s resilience to this attack.

### 2.4. Generation and Distribution of Symmetric Keys

A prerequisite for building a trusted environment is to ensure the confidentiality of transmitted data between clients. Due to the limited capabilities of IoT devices, symmetric rather than asymmetric cryptography algorithms are a better choice for encryption algorithms.

When symmetric cryptography is used, the same key is used for encryption and decryption. In such a situation, the problem of secure symmetric key distribution arises. The paper [34] compares different key distribution schemes and methods. Based on the results obtained, in the developed set of mechanisms for distributing the generated symmetric keys, the authors proposed using the Key Translation Centre (KTC) scheme with a direct method, which consists of generating the key by the requesting client and sending this key to the CE node for translation and transmission to the destination node.

Node N1, which intends to establish a connection with another node (e.g., N2), generates an AES-256 symmetric key SK using the TPM and encrypts it with the CE node’s public key. To perform this task, node N1 uses a special and secure key duplication procedure offered by the TPM. The resulting form of the key duplication can be transmitted via a non-secure link. From this secured form, only the TPM of the target node can retrieve the original SK. Based on the pID list, node N1 selects node N2 with which it wants to co-use the symmetric key. After encrypting the key and selecting the pID, node N1 sends a key translation request to the CE node (Figure 4). This request is preceded by activating the authentication mechanism between both nodes (Section 2.3 describes the authentication mechanism). With this approach, only authenticated and authorized devices will have symmetric keys to exchange data between each other.

After authenticating both parties and receiving a request from node N1, the CE node, using the resources of the local TPM, decrypts the received session key and encrypts it with the public key of node N2. Additionally, it generates a new sID for nodes N1 and N2, which will be used to identify the data exchange between these two nodes only. Node CE sends the translated key and the generated sID to node N2, and after receiving an acknowledgement of receipt from node N2, the CE node sends the sID session identifier to node N1.

At the end of this procedure, only nodes N1 and N2 have a common session identifier (sID) and a symmetric key (SK) to secure data exchange between these nodes.

### 2.5. Renewal of Symmetric Keys

An important issue regarding the security of cryptographic keys is their validity. Using the same key for encryption for too long increases the risk that a potential adversary will be able to break it. For this reason, each system should have criteria for assessing key security. These criteria may include the time elapsed since the key was generated, the number of encrypted messages, or the number of bytes encrypted.

In the proposed solution, where network security is managed centrally through the CE node, keeping track of the number of bytes encrypted by each session key is extremely difficult. For this reason, the best solution appears to be renewing the keys at specified time intervals. The session key lifetime is configurable and determined by the system administrator, depending on the operational context and risk level. Once the defined period expires, an automatic key renewal procedure is initiated to minimize potential key exposure. Whenever a node finds an expired key, it restarts the key distribution procedure described in Section 2.4. In the case of a master key, key renewal is done by running the ECDH protocol, during which the exchange of public values to establish the new key is encrypted using the old master key.

## 3. Results

### 3.1. Demonstrator

A three-dev network demonstrator was developed to test the proposed procedures. One acts as the CE, and the other is a client.

Each of the IoT nodes is running on a Raspberry Pi board. The configuration includes a hardware TPM 2.0 module to provide hardware support for cryptographic operations, while inter-device communication is via a LoRa communication module. A significant advantage of using LoRa is its transmission range, which can reach dozens of kilometers. In addition, each node has a BME 280 sensor installed to measure air temperature, pressure, and humidity, which will serve as a data source in the experiments. Figure 5 shows the hardware components of the system node, and Table 1 shows the specifications of the node components.

Although the demonstrator relies on the random number generator integrated into the TPM for convenience and consistency with the proposed trust architecture, the method is not limited to this source of entropy. In practical implementations, a hardware-based quantum random number generator (QRNG) can be employed to improve the entropy quality and mitigate potential weaknesses associated with TPM-based randomness. For instance, commercially available QRNG devices such as the ID Quantique Quantis IDQ250C3 or IDQ250C2 may be integrated into the system.

### 3.2. Demonstrator Node Description

Each node in the system stores the data necessary for secure operation in a networked environment in its local resources. These resources are of two types:Data stored on the TPM;Data stored in node storage.

Figure 6 shows the system resources stored on each node during network operation.

#### 3.2.1. Data Stored in TPM Resources

In addition to its key generation and data encryption functions, the TPM also provides a means of securely storing these keys. It acts as a safe from which retrieving a key in plaintext is impossible.

Each node in the network stores five cryptographic keys in the local TPM. These include the RSA private key (RK) and the RSA public key (RPK), which are used for the secure transmission of session keys (SKs). The TPM stores symmetric session keys in its resources. In addition, a BK for broadcast transmission and an LEK (Local Encryption Key) used for local file encryption are stored in the TPM.

Another essential element of the TPM is the PCR registers. These registers store the AF values for authentication procedures. Any change to the node’s hardware and software configuration will alter the AF value, preventing the device from authenticating itself to the network.

#### 3.2.2. Data Stored in Node Storage

Node storage stores data that are responsible for the network’s proper functioning. In the demonstrator, each node has two files:devices.info—a file storing the pID of the network nodes (e.g., N1) and their descriptions of these nodes;devices.ini—a file storing data about the nodes with which the connection has been established (the session key is shared). Each entry in this file contains the pID of the device and the sID assigned to the pair of nodes by the CE. The LEK protects this file.

In addition to the files described, the system has dedicated space for other files used during network operation. These include files containing MK_CE_ and the AF_CE_ value. The LEK protects both of these files. The CE node also stores all clients’ RSA public keys (RPK_Ni_).

### 3.3. Test Cases

Due to the limited availability of fully functional commercial hardware TPMs supporting the complete set of cryptographic operations required for this study, the demonstrator implementation partially relies on a software TPM environment. Specifically, the use of the software TPM was necessary to support symmetric encryption and decryption, as this functionality is not accessible through the currently available hardware interfaces.

Despite this limitation, the following components of the trust and security architecture are assumed to be handled by a hardware TPM in the target production environment:Creation and secure storage of each node’s trust structure;Generation and management of asymmetric and symmetric cryptographic keys;Secure random number generation;SHA-256 digest computation;HMAC generation;Signature creation and verification using RSA;RSA-based encryption and decryption.

To preserve the consistency of trust structures, all cryptographic elements were first generated within the TPM environment. When symmetric encryption or decryption was required during testing, the symmetric key was securely duplicated from the hardware TPM to the software TPM using standard TPM-supported procedures (TPM2_Duplicate and TPM2_Import). This hybrid approach allowed us to simulate real-world behavior while adhering as closely as possible to hardware-rooted trust architecture.

Each of the developed procedures was implemented and tested using the demonstrator. Figure 7 shows the console view during the initialization of the client node. First, the hardware TPM generates a pair of RSA keys, which are used to distribute the key between the different TPMs. The hardware TPM available in the experiments does not support symmetric key encryption, so a software TPM is used. Then, the software TPM generates RSA keys, which are used to transfer the key from the hardware TPM to the software TPM. Finally, a random number is generated (Connection Key), which is used as a seed to generate the symmetric key (Master Key).

The next important step in the demonstrator’s operation is attaching the node to the network. Figure 8 illustrates how this procedure works. In this stage, the actions described in Section 2.2 are performed. The one-time key generated by node N1 and inserted into the resources of node CE is highlighted in yellow.

The last test concerns the distribution of a symmetric key. Figure 9 shows the flow of this test. The procedure starts with the mutual authentication of the two nodes (red frame in Figure 9), as described in Section 2.3. Then, the node requesting the distribution (N1) generates a session key (SK) using the hardware TPM and encrypts it with the public key RPKCE. In the next step, node N1 sends it to the CE node for translation and transmission to the N2 node (green frame), as described in Section 2.4.

Table 2 shows the duration of each developed procedure on each network node. The following columns display the results for the CE node, the N1 node that joined the network and requested the translation of the key, and the N2 node, which was only the recipient. The bandwidth achieved during the tests was about 0.5 kbps. Despite the low throughput, the received times show that a better LoRa module offering higher throughput performed well.

The proposed solution minimizes network overhead by limiting the number of short, fixed-size frames exchanged during each protocol phase. During the network-joining procedure, a total of five frames are transmitted: two from the central entity (CE) to the joining node (N1), two in the opposite direction, and a single broadcast message to all nodes. With each frame limited to 255 bytes, this results in approximately 1275 bytes of data transferred per registration event.

The authentication procedure is highly efficient, requiring only one frame from the client to the CE and one frame in return. This results in 510 bytes of total transmission overhead per session. Since authentication occurs frequently during normal operation, its lightweight nature makes it well suited for bandwidth-constrained environments.

The key distribution process, which enables secure communication between two end nodes, involves a total of 11 frames. These include interactions between each node and the CE, as well as peer-to-peer exchanges. This results in an overhead of 2805 bytes per session. Although this phase generates more traffic, it is initiated only when a node explicitly requests to communicate with another node. As such, it does not continuously burden the network, and its impact is acceptable even in constrained IoT settings.

The summary is presented in Table 3, which shows the total network load (in bytes) for each protocol phase: node registration, authentication, and key distribution.

## 4. Discussion

Test scenarios were developed for each procedure to confirm the correctness of the operations. Thanks to long authentication factors, the implemented authentication mechanism is characterized by resistance to brute-force attacks comparable to AES-256 security. The procedure also eliminates the risk of compromising the client node regarding hardware configuration changes, thanks to storing authentication factors in CE resources. If a node is taken over and unauthorized changes are made to it, the AF value will change. It will not match that stored in the CE node’s resources, resulting in the node being unable to authenticate.

The key distribution procedure ensures the secure distribution of secret key (SKs) between client nodes. Using symmetric cryptography reduces the risk of key compromise, given the upcoming development of quantum computers. It is also resistant to the Shor algorithm. The cyclic renewal of the SKs additionally contributes to security. If the message’s sender identifies the key as expired, the key distribution procedure is restarted. Should a message be sent despite an expired key, it will be rejected by the recipient, which also controls the validity of the key.

The proposed communication procedures are resistant to Man-In-The-Middle attacks. Using the sID assigned by the CE node prevents unauthorized parties from identifying communication parties. In addition, the method is also resistant to network listening for outgoing messages from a single node. Each data transmission between different pairs of client nodes is tagged with a different sID.

Despite the known limitations associated with the use of TPM technology and the RSA algorithm in the context of the IoT, the approach proposed in this work remains justified and practically viable. The TPM provides hardware-based mechanisms for secure key storage and cryptographic operations, adhering to the FIPS 140 standard, which is a significant asset for building trusted computing environments. Although the RSA algorithm, used in the key distribution procedure, is formally considered deprecated, it remains one of the most widely adopted asymmetric cryptographic algorithms in existing implementations due to its compatibility and widespread acceptance in production systems. In the proposed model, these technologies were deliberately selected and applied—with full awareness of their limitations and practical strengths—to enable the development of a secure and implementable solution tailored to IoT environments.

The proposed solution is designed for environments with limited bandwidth and a stable number of connected devices, where the frequent addition or removal of nodes is not anticipated. It applies to scenarios such as remote monitoring, infrastructure edge deployments, or sensor networks operating under constrained conditions. Although the key distribution process may appear time-consuming (as shown in Table 2), it is important to note that it occurs only on demand and only once per communicating node pair. In contrast, the authentication mechanism is lightweight, fast, and used repeatedly during system operation. Furthermore, during the node registration procedure, the updated list of registered nodes is distributed using a single broadcast message, which prevents the system from being affected by the number of devices in the network. As a result, the proposed architecture remains efficient and scalable within its target context. Future research needs to refine the communication mechanisms by developing a lightweight communication protocol for the LoRa interface. It is also essential to harden network devices, increasing their resistance to physical interference.

## Figures and Tables

**Figure 1 sensors-25-03881-f001:**
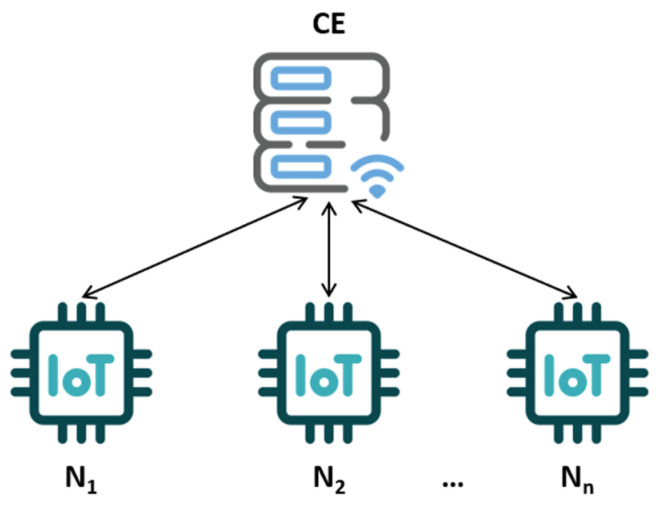
System demonstrator architecture.

**Figure 2 sensors-25-03881-f002:**
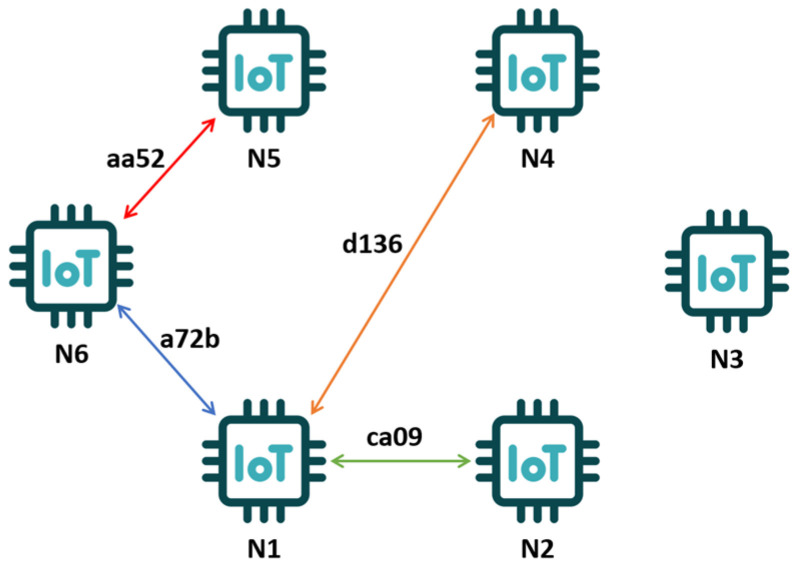
Example of how to identify data exchange links.

**Figure 3 sensors-25-03881-f003:**
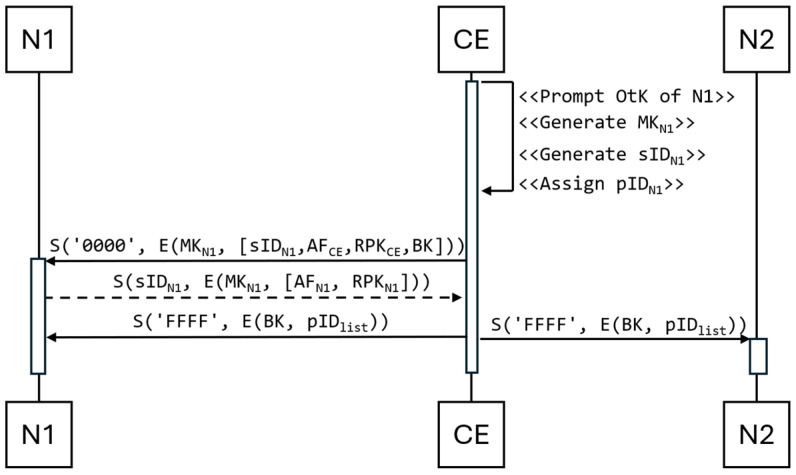
Diagram of the sequence for attaching a client to the network.

**Figure 4 sensors-25-03881-f004:**
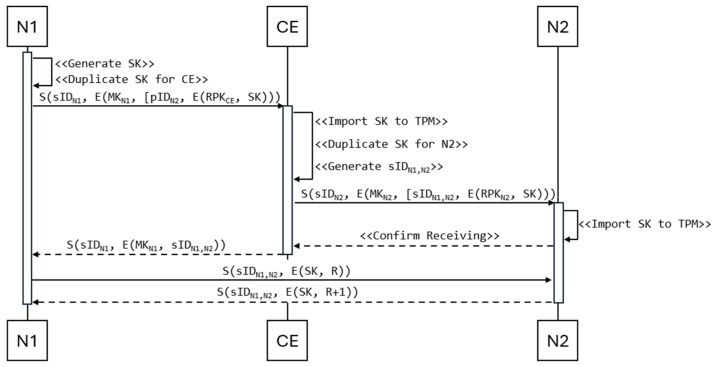
Symmetric key distribution sequence diagram.

**Figure 5 sensors-25-03881-f005:**
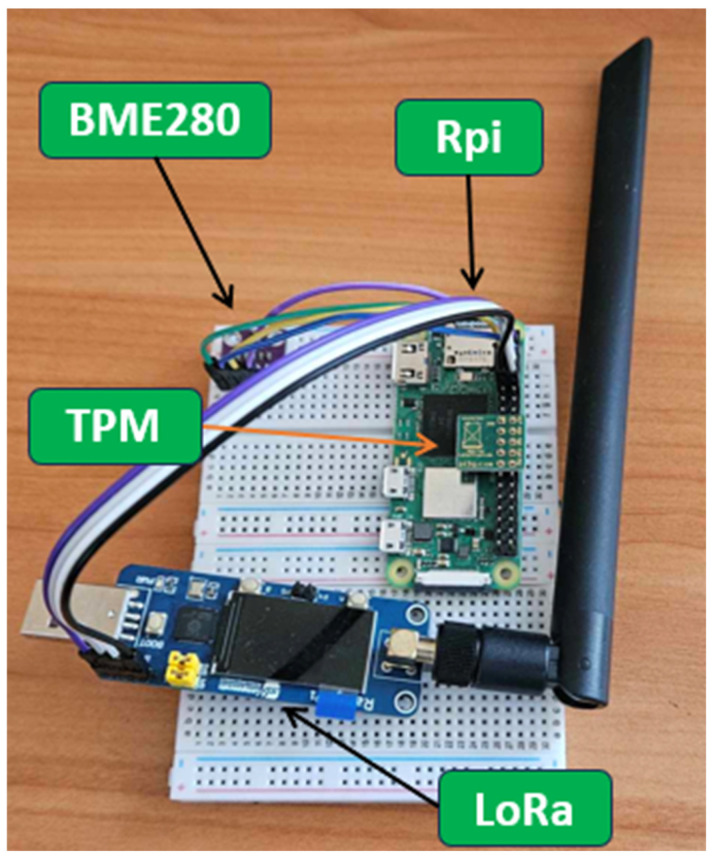
Demonstrator node view.

**Figure 6 sensors-25-03881-f006:**
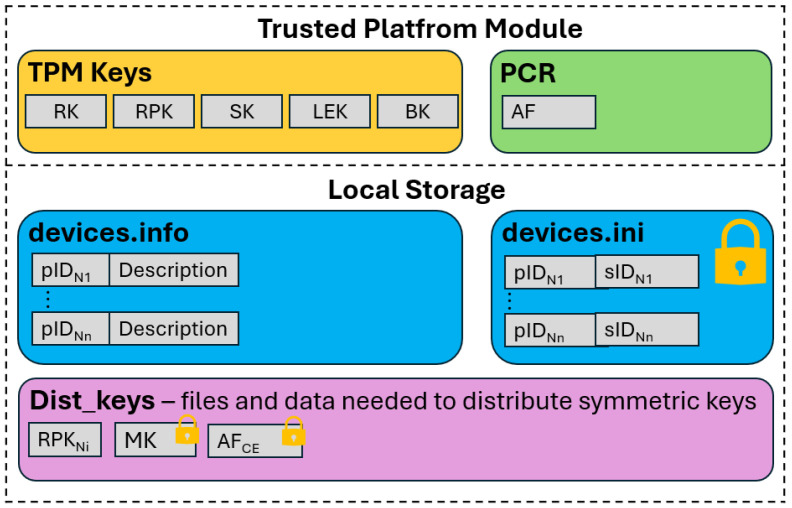
Data storage locations in system node resources.

**Figure 7 sensors-25-03881-f007:**
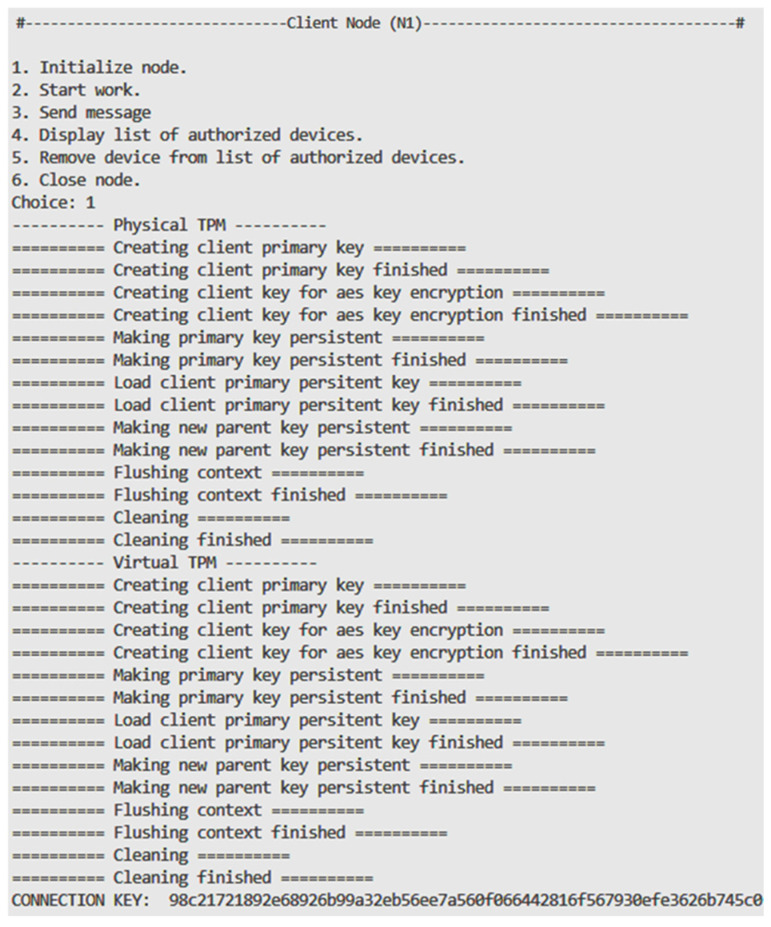
Console view of the N1 node during the initialization procedure.

**Figure 8 sensors-25-03881-f008:**
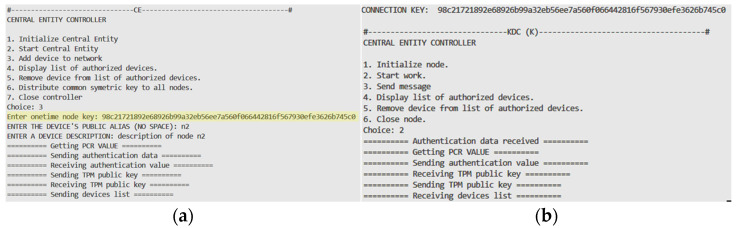
Console view of nodes N1 (**a**) and CE (**b**) when connecting to the network.

**Figure 9 sensors-25-03881-f009:**
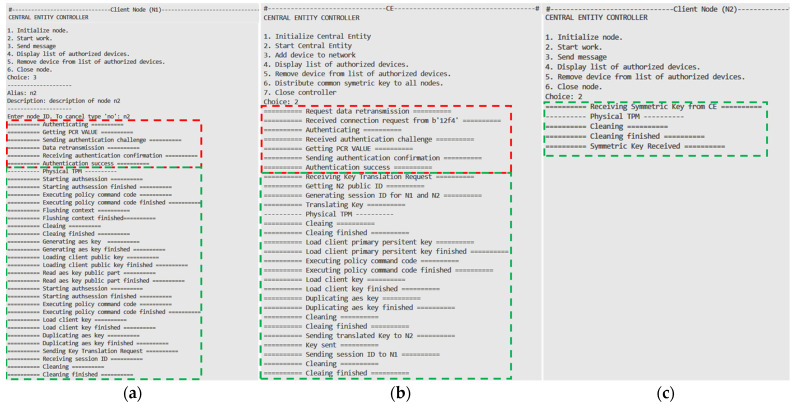
Console view of nodes N1 (**a**), CE (**b**), and N2 (**c**) during key distribution.

**Table 1 sensors-25-03881-t001:** Technical parameters of the node components.

Module	Model	Properties
Controller	Raspberry Pi Zero 2W	RAM: 512 MB Clocking: 1 GHz Cores: 4
Cryptographic module	LetsTrust TPM 2.0	Interface: SPI Voltage: 3.3 V Chip: Infineon Optiga SLB 9670 Properties: True Hardware Random Number Generator (FIPS 140-2)
Communication module	LoRa SB Component	Interface: UART Voltage: 5 V Chip: Semtech SX1262 Bandwidth: 868 MHz
Sensor	BME 280	Interface: I^2^C Voltage: 3.3 V Measurements: temperature, humidity, and pressure

**Table 2 sensors-25-03881-t002:** Average duration of the procedures at different nodes.

Action	CE Node [s]	N1 Node Requesting the Node [s]	N2 Node Receiving the Node [s]
Joining the network	41.9	35.6	1.29
Key distribution	47.5	53.7	1.34
Authentication	12.48	12.48	12.48

**Table 3 sensors-25-03881-t003:** Network load by data flow.

Procedure	Data Flow	Byte Transferred [B]	Sum [B]
Joining the network	CE → N1	510	1275
	N1 → CE	510	
	CE → broadcast	255	
Authentication	N1 → CE	255	510
	CE → N1	255	
Key distribution	N1 → CE	510	2805
	CE → N2	510	
	N2 → CE	255	
	CE → N1	510	
	N1 → N2	510	
	N2 → N1	510	

## Data Availability

Data is not publicly available due to the classified nature of the work.

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
