# Peer review of "Procedures for Building a Secure Environment in IoT Networks Using the LoRa Interface"

_sensors, 2025, doi:10.3390/s25133881_

Round 1
Reviewer 1 Report
Comments and Suggestions for Authors
The article presents a scheme for building a secure environment in IoT networks, using LoRa interface. It is adapted to work in an untrusted environment and allows one to create a trusted environment, starting from the very beginning of the network. It is impossible to compromise transmitted data or unauthorized devices. The scheme proposed by the authors assumes a mechanism for authenticating IoT nodes, a procedure for registering new devices, and a method for distributing symmetric cryptographic keys. In addition, a new method for identifying network devices was proposed, which makes it impossible for unauthorized parties to identify the sender and recipient of a message. The protocol for authenticating devices uses Platform Configuration Registers (PCR) in its work and operates in accordance with the Proof-of-Knowledge concept.
The authors state that the proposed scheme is resistant to Man-in-the-Middle attacks. The use of session identifiers of nodes prevents the identification of unauthorized network participants. In addition, the scheme is resistant to network eavesdropping of outgoing messages from one node.
Comments and questions:
The article does not provide a full overview of relevant sources on the topic of creating secure (trusted) environments in the IoT. Only the Distributed Ledger Technology (DLT) scheme is mentioned. Are there probably already existing approaches to creating a secure environment in the IoT that are based on cryptography and other mechanisms? What are their disadvantages and what is the advantage of the proposed solution compared to similar ones?
The Introduction and Conclusion sections note that the main disadvantage of asymmetric cryptography schemes is the risk of compromising keys using quantum computers and Shor's algorithm. However, asymmetric cryptography is used during the procedure of exchanging Session Key (SK) and Session Identifiers (sIDs) to establish a secure connection between nodes (Figure 4, the Session Key is encrypted with the public key of another node and CE for its transmission over the network). Will this fact further compromise the entire proposed scheme?
How are network devices (nodes) protected from software hacking and compromise attacks (without hardware modification and modification of the equipment configuration)?
The work does not include an analysis of the resource consumption of network nodes, which should show how much the proposed scheme additionally loads the network.
The proposed mechanism for implementing a secure environment does not imply automatic addition of new devices during network operation. Since each device must be registered by the network administrator in the CE (Central Entity) by adding a one-time key (OtK) of the device.
Author Response
Thank you very much for taking the time to review this manuscript. Please find the detailed responses below and corresponding corrections/corrections highlighted in blue in the resubmitted file.
Comments 1:
The article does not provide a full overview of relevant sources on the topic of creating secure (trusted) environments in IoT. Only the Distributed Ledger Technology (DLT) scheme is mentioned. Are there probably already existing approaches to creating a secure environment in the IoT that are based on cryptography and other mechanisms? What are their disadvantages and what is the advantage of the proposed solution compared to similar ones?
Response 1:
Thank you for pointing this out. We have extended our literature review to include additional relevant solutions. We also highlighted the limitations of these approaches and emphasised the advantages of our proposed method in comparison. Mention exactly where in the revised manuscript this change can be found - page 3, lines 102-141 and page 3, lines 152-166.
Comments 2:
The Introduction and Conclusion sections note that the main disadvantage of asymmetric cryptography schemes is the risk of compromising keys using quantum computers and Shor's algorithm. However, asymmetric cryptography is used during the procedure of exchanging Session Key (SK) and Session Identifiers (sIDs) to establish a secure connection between nodes (Figure 4, the Session Key is encrypted with the public key of another node and CE for its transmission over the network). Will this fact further compromise the entire proposed scheme?
Response 2:
Thank you for this insightful comment. We agree that the use of asymmetric cryptography poses a potential risk in the context of future quantum computing capabilities, particularly due to Shor's algorithm, which may compromise public-key cryptosystems. As noted in the manuscript, this is considered one of the limitations of the proposed approach. However, it is essential to note that asymmetric cryptography in our solution is employed exclusively during the initial exchange of the Session Key (SK) and Session Identifiers (sIDs). These elements are then used for subsequent secure communication between nodes. This means that any quantum-related risk is limited to the initialisation phase and does not impact the overall security of ongoing communication.
Furthermore, the proposed architecture can be adapted in the future to incorporate post-quantum key exchange mechanisms without requiring significant changes to the rest of the system. Given the current state of quantum computing, we believe the proposed approach remains both practical and secure. Nevertheless, this issue is addressed in the "Discussion" section - page 166, lines 558-568.
Comments 3:
How are network devices (nodes) protected from software hacking and compromise attacks (without hardware modification and modification of the equipment configuration)?
Response 3:
Thank you for this question. It is important to note that the scope of this article focuses on network-level procedures for trust establishment, node registration, and authentication in IoE environments rather than on-device hardening or operating system-level security mechanisms. The proposed solution assumes the use of standard security features available in devices equipped with a TPM module. It should also be clarified that cryptographic keys are not permanently stored within the TPM. Instead, they are regenerated within the secure environment of the TPM using encrypted seeds that are previously imported into the module. This approach ensures the confidentiality and integrity of the keys without requiring them to be persistently stored in device memory, thereby reducing the risk of key exposure or compromise. The TPM is used in our model to support secure encryption and signing operations during the registration and key exchange procedures. In response, we have added a paragraph to clarify the scope of the considered adversarial capabilities in the Introduction - page 4, lines 177-186.
Comments 4:
The work does not include an analysis of the resource consumption of network nodes, which should show how much the proposed scheme additionally loads the network.
Response 4:
Thank you for this valuable remark. We agree that analysing resource consumption is an important aspect when evaluating the suitability of a protocol for constrained IoT environments. Although a detailed power or CPU usage profile of each node was not the primary focus of this work, we addressed the issue of network load by quantifying the number of messages and the size of transmitted data for each protocol phase (see Table 3). These measurements offer an indirect yet practical insight into the resource burden placed on the network.
It is also worth emphasising that the computational operations involved in the proposed authentication mechanism are lightweight and do not require complex cryptographic primitives such as public key operations during normal communication. It makes the scheme feasible for deployment on low-power devices. In future work, we plan to complement the current evaluation with measurements of CPU usage, memory footprint, and energy consumption to provide a more comprehensive view of system performance. The current results related to network usage and message load are summarised at the end of Section 3.3 - page 15, lines 518-536.
Commnents 5:
The proposed mechanism for implementing a secure environment does not imply automatic addition of new devices during network operation. Since the network administrator must register each device in the CE (Central Entity) by adding a one-time key (OtK) of the device.
Response 5:
It is a shortcoming that requires the system administrator's attention. However, this approach allows tighter control of the data entered into the system in the first step, which conditions the registration of only trusted devices on the network. It is worth noting that this is a one-off activity. Currently, this is done manually. Still, it is possible to successfully adapt the system to enter verified data in a batch manner at this stage, which will undoubtedly make the system easier to use. See a footnote on page 7, line 261.

Reviewer 2 Report
Comments and Suggestions for Authors
This paper proposes a comprehensive framework for securing resource-constrained IoT networks using LoRa communication. The solution addresses IoT-specific challenges like low bandwidth and computational constraints while emphasizing quantum-resistant symmetric cryptography.
(1) Security Analysis Gaps: The authentication mechanism (Section 2.3) claims resistance to brute-force attacks due to the 32-byte AF value but lacks formal proof or reference to cryptographic standards. Provide a security proof (e.g., reduction to hash function security) or cite established frameworks (e.g., indistinguishability under chosen-plaintext attacks).
(2) Threat Model Omission: The paper does not explicitly define adversarial capabilities (e.g., physical tampering, network eavesdropping). Clarify which threats the system mitigates (e.g., impersonation via PCR changes) and which are out of scope (e.g., side-channel attacks on TPM).
(3) LoRa-Specific Limitations: While sID/pID addressing hides device identities (Section 2.2), LoRa’s long-range nature raises privacy risks (e.g., location tracking via signal triangulation). Discuss how the scheme counteracts physical-layer vulnerabilities unique to LoRa.
(4) Performance Scalability Concerns: Table 2 shows key distribution takes ~53 seconds for requesting nodes. For large-scale IoT deployments, this latency may be prohibitive. Evaluate scalability beyond three nodes and propose optimizations (e.g., batch key distribution).
(5) Comparative Evaluation: The solution is compared only to generic key distribution schemes (Section 1). Benchmark against state-of-the-art IoT protocols (e.g., OSCORE, DTLS) in terms of energy consumption, bandwidth overhead, and security guarantees.
(6) Demonstrator Implementation Details: The "software TPM" workaround for symmetric encryption (Section 3.3) undermines hardware-rooted trust claims. Justify this design choice or revise the implementation to use hardware TPM exclusively.
(7) Insufficient references: Lack of recent cryptography-related articles that need to be cited in the context of the introduction. (e.g. Chaotic Encryption Using Hybrid Evolution Cellular Automata and 4D Modulation for Data Centers, Privacy Image Secrecy Scheme Based on Chaos-Driven Fractal Sorting Matrix and Fibonacci Q-Matrix, Lightweight Image Encryption Algorithm Using 4D-NDS: Compound Dynamic Diffusion and Single-Round Efficiency)
(8) Quantum Attack Resilience: The emphasis on symmetric cryptography’s quantum resistance (Section 1) overlooks practical threats (e.g., Grover’s algorithm). Quantify security margins by specifying recommended key lifetimes.
Author Response
Thank you very much for taking the time to review this manuscript. Please find the detailed responses below and corresponding corrections/corrections highlighted in blue in the resubmitted file.
Comments 1:
Security Analysis Gaps: The authentication mechanism (Section 2.3) claims resistance to brute-force attacks due to the 32-byte AF value but lacks formal proof or reference to cryptographic standards. Provide security proof (e.g., reduction to hash function security) or cite established frameworks (e.g., indistinguishability under chosen-plaintext attacks).
Response 1:
Thank you for this comment. It is a valid point, and it is indeed necessary to provide a more detailed explanation of the security properties of the proposed protocol. Therefore, we have revised and expanded the security analysis, which is now located at the end of Section 2.3 - page 9, line 320-341.
Comments 2:
Threat Model Omission: The paper does not explicitly define adversarial capabilities (e.g., physical tampering, network eavesdropping). Clarify which threats the system mitigates (e.g., impersonation via PCR changes) and which are out of scope (e.g., side-channel attacks on TPM).
Response 2:
Thank you for this valuable comment. We agree that a clear definition of the threat model is important for evaluating the security of the proposed solution. In response, we have added a paragraph to clarify the scope of the considered adversarial capabilities in the Introduction. This work focuses on network-level procedures, such as trust establishment and mutual authentication, rather than system-level or hardware hardening. Therefore, we assume a network adversary who can intercept, modify, or replay messages and attempt to impersonate legitimate nodes. The protocol provides resistance to such attacks, including attempts to forge authentication data or disrupt the establishment of sessions. However, we explicitly state that physical tampering, side-channel attacks, and firmware modification are outside the scope, as the integrity of the TPM and the operating environment is assumed to be trusted - line 4, lines 177-186.
Comments 3:
LoRa-Specific Limitations: While sID/pID addressing hides device identities (Section 2.2), LoRa's long-range nature raises privacy risks (e.g., location tracking via signal triangulation). Discuss how the scheme counteracts physical-layer vulnerabilities unique to LoRa.
Response 3:
The study presented did not focus on mitigating threats to the LoRaWAN physical layer. However, it does interact with these threats to some extent. The introduction of sID and pID, together with the authentication Factor (AF) parameter, eliminates Man-in-the-Middle attacks that could result from successful sniffing and key extraction attacks. A common vulnerability in LoRaWan systems is the storage of transmission security keys at the physical layer in plaintext. These keys can be stored in an implicit form protected by the mechanisms of the TPM and thus prevent key extraction.
Jamming and energy attacks are difficult to thwart. The wormhole attack is based on successful sniffing and jamming attacks. The proposed solution does not support immunity against these attacks.
In this article, the last paragraph in section 2.3 is devoted to these issues page 9, lines 342-358.
Comments 4:
Performance Scalability Concerns: Table 2 shows key distribution takes ~53 seconds for requesting nodes. For large-scale IoT deployments, this latency may be prohibitive. Evaluate scalability beyond three nodes and propose optimisations (e.g., batch key distribution).
Response 4:
Thank you for bringing this important concern to our attention. We want to clarify that the proposed solution is designed for environments characterised by limited bandwidth availability and a stable node population, where frequent addition or removal of devices is not expected. This includes scenarios such as remote monitoring systems, infrastructure edge installations, or sensor networks in constrained environments.
It is also important to emphasise that the registration procedure is performed only once per device and the key distribution procedure occurs only on demand and only once per communicating node pair. Therefore, although the key distribution may appear time-consuming in Table 2, its impact on overall system performance is minimal in the long term.
Moreover, during the registration process, the updated list of registered nodes is sent via a single broadcast message, ensuring that the distribution of this information to all network participants does not scale linearly with the number of devices. As a result, the final stage of registration adds negligible overhead, even in larger setups. Nevertheless, we have added a dedicated paragraph to the conclusion section that addresses the scalability aspect of the proposed solution - page 17, lines 569-579.
Comments 5:
Comparative Evaluation: The solution is compared only to generic key distribution schemes (Section 1). Benchmark against state-of-the-art IoT protocols (e.g., OSCORE, DTLS) in terms of energy consumption, bandwidth overhead, and security guarantees.
Response 5:
Thank you for highlighting the importance of benchmarking the proposed solution against established IoT security protocols. In response to your suggestion, we have added a comparative discussion referencing state-of-the-art solutions such as OSCORE [RFC 8613] and DTLS [RFC 9147] in the revised version of the manuscript. These protocols provide strong end-to-end security guarantees but rely on pre-shared keys or certificates and may introduce notable communication and computational overhead, particularly in constrained environments. In contrast, our method focuses on initial trust establishment without prior knowledge and offers a lightweight exchange model suitable for low-bandwidth, low-power scenarios. Although the current work does not include detailed benchmark results on energy consumption and bandwidth, we have clarified these comparative aspects in the introduction section - page 3, lines 127-141.
Comments 6:
Demonstrator Implementation Details: The "software TPM" workaround for symmetric encryption (Section 3.3) undermines hardware-rooted trust claims. Justify this design choice or revise the implementation to use hardware TPM exclusively.
Response 6:
A significant point. The reason for this approach is the lack of access to a fully functional TPM hardware module in the commercial market. The authors of the study do not have access to such a hardware TPM module. In this situation, We assumed that the hardware TPM module will support the following solution components:
- creation and storage of each node's trust structure;
- generation and key storage of asymmetric and symmetric keys;
- generation of random numbers;
- determination of SHA digests;
- determining HMAC digests (Sebastian, I don't know if you use this);
- signing and verifying signatures using RSA;
- RSA encryption and decryption.
We used the software TPM module for symmetric encryption and decryption. Before using the software TPM module, a trust structure was generated on the module. When symmetric encryption or decryption was needed, a symmetric key from the hardware TPM module was duplicated into the software TPM module using the appropriate TPM-assisted secure procedure. We have included a detailed explanation of this design choice at the beginning of the Test Cases section to clarify the reasons for using a software TPM module in the current implementation - page 13, lines 466-487.
Comments 7:
Insufficient references: Lack of recent cryptography-related articles that need to be cited in the context of the introduction. (e.g., Image privacy protection scheme based on high-quality reconstruction DCT compression and nonlinear dynamics, Chaotic Encryption Using Hybrid Evolution Cellular Automata and 4D Modulation for Data Centers, Privacy Image Secrecy Scheme Based on Chaos-Driven Fractal Sorting Matrix and Fibonacci Q-Matrix, Lightweight Image Encryption Algorithm Using 4D-NDS: Compound Dynamic Diffusion and Single-Round Efficiency).
Response 7:
Thank you for this valuable suggestion. We agree that including references to recent cryptography-related publications strengthens the context of our contribution. In response to your comment, we have cited all of the mentioned works in the revised Introduction section. These papers have been used to highlight current trends in lightweight and chaos-based cryptographic techniques, which align with the design philosophy of our proposed authentication and trust model. The relevant citations have been integrated into the introductory discussion to emphasise the connection between emerging cryptographic strategies and the practical requirements of secure communication in constrained environments - page 3, lines 111-126.
Comments 8:
Quantum Attack Resilience: The emphasis on symmetric cryptography's quantum resistance (Section 1) overlooks practical threats (e.g., Grover's algorithm). Quantify security margins by specifying recommended key lifetimes.
Response 8:
Thank you for this valuable observation. In response, we have expanded the discussion of quantum threats and key lifetime management in the Introduction and Section 2.5. We now explicitly reference Grover's algorithm as a practical quantum threat and explain its impact on the security levels of symmetric keys. To address this, we recommend using 256-bit symmetric keys (e.g., AES-256), which provide an effective 128-bit security level in the presence of quantum adversaries. Additionally, we have clarified that session key lifetimes should be strictly limited in accordance with NIST SP 800-57 Part 1 Rev. 5 guidelines. Specifically, we indicate that keys used for short-term protection should have a usage period of one day or less, and that these durations remain valid even in the post-quantum context, provided that the key size is appropriately increased. Moreover, we emphasise that session key lifetimes are configurable within the proposed system and that an automated key renewal procedure is triggered once the defined validity period expires - page 2, lines 58-77 and page 11, lines 402-405.

Reviewer 3 Report
Comments and Suggestions for Authors
the paper propose a solution to increase the resilience of IoE based systems.
the surveilled literature is good
the conclusions match the rest of the paper
using TPM in IoE is not new but their approaches is
yet, we are into an state level asymmetrical war at the cybersecurity level from 2004
as a result, especially in the new context of Russia menace over the UE is expected that military thinking and solutions must be adopted in each part of the complex hardware and system that is specific to an informational security society
the military thinking states that.
1..what I use must be under my control
2. if I use third party solutions an deep security SWOT must be done before adoption
and the list may continue
anyway in this context some observations arise
1. the use of the TPM is obsolete for IoE solutions due to its security problems that arise just before it begin to be used a long time ago (for a small part of its problem you mat consult by example gr0k). Also the random generator have some limitation
2. the use of RSA witch is already deprecated - for example see the attached paper or just consult the internet using AI or not
even more advances solutions such as the one presented in:
https://ieeexplore.ieee.org/abstract/document/9906050
still have some problems
In this context probably it will be a good idea to add some comment concerning these problems and how your solution mitigate them
one directly related to the paper where if you take a look you will see that mainly my observations was mainly favorable with some observation at the final that they must add some information needed to defend the proposed paper in front of the many problems that TPM or RSA use have. This was needed to convince the reader that this approach is still viable in the current context! In UE due to MS influence using TPM is still considered as a good solution so from this point of view the paper is publishable because their solution differs from the rest of proposals of using TPM in IoE.
the other concerns the research direction of the authors and so called criticisms are in fact suggestions that may help them in order to shift in researching more secure approaches because when I check their background I see that they are from a military academy so the used security standards I expect to be a little higher than the civil universities.
The observation for editor was generated by the fact that the part of cryptanalysis of the new proposed method is a little bit exhaustive and this is a stile of an bachelor student
in conclusion the paper is OK

Author Response
Thank you very much for taking the time to review this manuscript. Please find the detailed responses below and corresponding corrections/corrections highlighted in blue in the resubmitted file.
Comments 1:
What I use must be under my control.
Response 1:
We fully agree with the reviewer's opinion.
Comments 2:
If I use third-party solutions, a deep security SWOT analysis must be done before adoption, and the list may continue. In this context, some observations arise.
Response 2:
We fully agree with the reviewer's opinion.
Comments 3:
The use of the TPM is obsolete for IoE solutions due to its security problems that arisen just before it begin to be used a long time ago (for a small part of its problem you mat consult by example gr0k). Also, the random generator have some limitation .
Response 3:
In the prepared demonstrator, it was convenient to use the random generator of the hardware TPM module. If the proposed method is implemented, it is possible to use a quantum random number generator such as the Quantis QRNG IDQ250C3 or IDQ250C2. One of the co-authors has performed such experiments with a previous version of this type of generator. A corresponding note has been added to Section 3.1 of the manuscript - page 11, lines 421-427.
Comments 4:
The use of RSA witch is already deprecated - for example see the attached paper or just consult the internet using AI or not even more advances solutions such as the one presented in: https://ieeexplore.ieee.org/abstract/document/9906050 still have some problems. In this context probably it will be a good idea to add some comment concerning these problems and how your solution mitigate them one directly related to the paper where if you take a look you will see that mainly my observations was mainly favorable with some observation at the final that they must add some information needed to defend the proposed paper in front of the many problems that TPM or RSA use have. This was needed to convince the reader that this approach is still viable in the current context! In UE due to MS influence using TPM is still considered as a good solution so from this point of view the paper is publishable because their solution differs from the rest of proposals of using TPM in IoE.
Response 4:
Thank you for pointing this out. We fully acknowledge that RSA is considered deprecated and that the use of TPM may be viewed as obsolete in the context of modern cryptographic and security standards. To address this, we have included a detailed explanation in the "Discussion" section, where we clarify that we are aware of the limitations and drawbacks associated with these technologies. However, it is important to emphasise that both RSA and TPM are still widely deployed in numerous real-world systems, particularly in commercial environments where compatibility, standardisation, and certification (e.g., FIPS 140 compliance) play a critical role. The decision to incorporate these technologies into our model was therefore made consciously, considering their practical relevance, availability in existing hardware platforms, and broad support across different system implementations. While more modern alternatives exist, our approach demonstrates that, when applied appropriately, RSA and TPM can still provide a viable and secure foundation for key distribution and trust establishment in IoT scenarios - page 13, lines 558-568.
Comments 5:
The other concerns the research direction of the authors and so-called criticisms are, in fact, suggestions that may help them to shift in researching more secure approaches because when I check their background, I see that they are from a military academy, so they used security standards I expect to be a little higher than the civil universities.
Response 5:
The authors fully agree with this suggestion.

Round 2
Reviewer 2 Report
Comments and Suggestions for Authors
I have no further comments.